# Cellular advective-diffusion drives the emergence of bacterial surface colonization patterns and heterogeneity

Tamara Rossy [1], Carey D. Nadell [2] & Alexandre Persat [1]

Microorganisms navigate and divide on surfaces to form multicellular structures called biofilms, the most widespread survival strategy found in the bacterial world. One common assumption is that cellular components guide the spatial architecture and arrangement of multiple species in a biofilm. However, bacteria must contend with mechanical forces generated through contact with surfaces and under fluid flow, whose contributions to colonization patterns are poorly understood. Here, we show how the balance between motility and flow promotes the emergence of morphological patterns in *Caulobacter crescentus* biofilms. By modeling transport of single cells by flow and Brownian-like swimming, we show that the emergence of these patterns is guided by an effective Péclet number. By analogy with transport phenomena we show that, counter-intuitively, fluid flow represses mixing of distinct clonal lineages, thereby affecting the interaction landscapes between biofilm-dwelling bacteria. This demonstrates that hydrodynamics influence species interaction and evolution within surface-associated communities.

[1] Institute of Bioengineering and Global Health Institute, School of Life Sciences, École Polytechnique Fédérale de Lausanne, Lausanne 1015, Switzerland.
[2] Department of Biological Sciences, Dartmouth, Hanover, NH 03755, USA. Correspondence and requests for materials should be addressed to C.D.N. (email: carey.d.nadell@dartmouth.edu) or to A.P. (email: alexandre.persat@epfl.ch)

Biofilm formation is among the most important and widespread survival strategies found in the bacterial world[1]. This process occurs when microorganisms navigate and attach to surfaces, embedding themselves and their progeny in a matrix of extracellular polymeric substances (EPS) that confer multiple competitive advantages. Owing to the adhesive properties of the EPS matrix, biofilms help bacteria stably reside at surfaces with access to metabolic resources. This multicellular lifestyle also provides cells with a physical shelter, reducing population erosion in flow environments[2], increasing resistance to chemical stressors such as antibiotics[3], and limiting the invasion of bacterial and viral competitors within the community[4,5].

The spatial structure of biofilms is heterogeneous and dynamic[6]. Natural biofilms are thought to commonly include multiple strains and species that can be organized in a variety of three-dimensional patterns. For instance, dental plaque biofilms comprise multiple genera enclosed in a single structure with varying shape and clonal density at different spatial scales[7]. The spatial arrangements of such multi-species consortia can dramatically impact evolution of cell–cell interactions, and vice versa[6]. The expression of social phenotypes leads to the constant remodeling of biofilms: some strains are killed or constrained to regions with poor nutrient availability, while others may gain access to nutrient-rich space[8].

During surface colonization, bacteria must contend with a wide range of mechanical forces generated in contact with interfaces under fluid flow[9]. Despite their importance in the establishment of biofilms in the natural environment, their contributions to colonization patterns and biofilm architecture are not well understood. As one might suspect, hydrodynamic forces can disrupt biofilms, promoting removal of biomass from surfaces[10]. Less intuitively, fluid flow can promote unusual biofilm structures such as streamers, leading to sudden clogging of fluidic systems[11], affect the cellular organization of single cells within the biofilm[12] or alter the evolutionary dynamics of matrix secretion[13].

Bacterial motility plays a key role during early biofilm growth, for example, as single cells increase their rate of encounter with surfaces by swimming[14–17]. However, the joint roles of fluid flow, motility and bacterial surface interactions have only just begun to receive attention[18]. Given the importance of fluid flow in remodeling biofilms and in transporting planktonic cells or aggregates, we anticipate that such forces also modulate spatial organization of surface associated bacterial collectives on many scales.

Most microbes have evolved cellular components optimizing their interactions with surfaces[3,9,19]. C. crescentus is particularly well-adapted to life on surfaces under flow: a polar stalk and adhesive holdfast confer strong attachment, and its curved morphology promotes biofilm formation in flow[20,21]. During the process of growth on surfaces, C. crescentus mother cells asymmetrically divide into a nonmotile stalked cell that stays put on the surface and a daughter swarmer cell that may either attach to the surface of be carried by the flow[22]. The characteristic curved shape of C. crescentus promotes local surface colonization by reorienting the body of sessile mother cells in the direction of the flow, so that the piliated pole of the daughter cell is close to the surface. Relative to mutants with straight cell shape, this process accelerates accumulation of biomass near the founder cell, leading to the formation of clonal microcolonies[14,20,23,24]. During sessile division in flow, daughter swarmer cells may either attach immediately downstream of their mother cell or explore the surrounding fluid to later reattach. The former depends on cell shape while the latter must depend on fluid transport mechanisms and cell motility. The relative importance of these surface colonization modes will, we predict, dramatically influence the basal architecture and cell lineage structure of nascent biofilm populations.

Here, we sought to answer how hydrodynamic forces affect C. crescentus biofilm architecture and spatial lineage structure. Using microfluidics and fluorescence microscopy, we explored how the intensity of transport by flow could modulate patterns of surface occupation. Our results indicate that increasingly fast fluid flow shifts surface occupation away from flagellum-driven exploration, and contributes to the formation of larger, more segregated colonies. Using insights from mass transport phenomena, we propose a model based on diffusion and advection describing how hydrodynamics influence initial surface colonization. Finally, we demonstrate that the balance of between flow transport and swimming of planktonic cells strongly modulates the spatial organization of distinct bacterial clones, thereby driving biofilm heterogeneity, which in turn may impact the evolution of social phenotypes.

## Results

**Flow modulates bacterial surface colonization patterns**. We initially sought to investigate the contributions of fluid flow to surface colonization patterns. We first grew C. crescentus biofilms in different hydrodynamic conditions by exposing surface-associated cells to controlled flow in microfluidic channels. We observed striking differences in morphologies in the emerging sessile populations as a function of flow speed. In relatively weak flow (2 mm s$^{-1}$), C. crescentus rapidly colonized the surface of the channel without forming well-defined colonies (Fig. 1a). In contrast, spatial patterns of colonization emerged in strong flow (27 mm s$^{-1}$), where biofilms grew into sparse, dense microcolonies (Fig. 1b). Surface occupation dramatically dropped for growth at mean fluid velocity higher than 4 mm s$^{-1}$ (Fig. 1c versus Fig. 1d–e, and 1f). Surface colonization was also found to be faster in weak flow than in strong flow (Fig. 1g). Visualization at higher spatial resolution highlights the presence of many isolated single cells in intermediate flow (Fig. 1d), which are absent in stronger flows (Fig. 1e). As a result, clusters are generally small in weak to intermediate flow (median cluster area <40 μm$^2$), in comparison to strong flows (median cluster area > 100 μm$^2$) (Fig. 1h). Thus, flow promotes the emergence of multicellular patch-like patterns at the channel surface but slows down surface occupation.

**Swimming motility promotes surface colonization**. During asymmetric division, C. crescentus releases unattached progeny into the fluid bulk (Fig. 2a). By visualizing surface colonization dynamically, we observed that, in strong flow, biofilms develop from single founder cells (Supplementary Movie 1). Conversely, in weak flow, new founder cells frequently attach to the surface, speeding up the overall rate of colonization and homogenizing surface coverage (Supplementary Movie 2 and Fig. 1g). We thus suspected that the relative contribution of random spatial exploration by swimming motility, and flow transport may enable reattachment. To demonstrate this, we abolished swimming motility by deleting the flagellar gene flgE. For this mutant, flow is the dominant transport mechanism of swarmer cells. Figure 2b shows a comparison between biofilms formed by wild-type (WT) and flgE$^-$ in weak flow. In contrast to WT, flagellum-less cells colonize the surface into patch-like patterns and single isolated cells are rare, reminiscent of the patterns observed with WT in intermediate flow (Fig. 1d). While the obvious effect of a flgE$^-$ mutation is loss of motility, recent work indicates that these can affect the regulation of holdfast synthesis[24,25]. We thus tested whether the distinct biofilm patterns observed between WT and flgE$^-$ cells could indeed be attributed to differences in motility rather than in adhesive properties. We already know that the proportion of swarmer cells attaching to the surface immediately

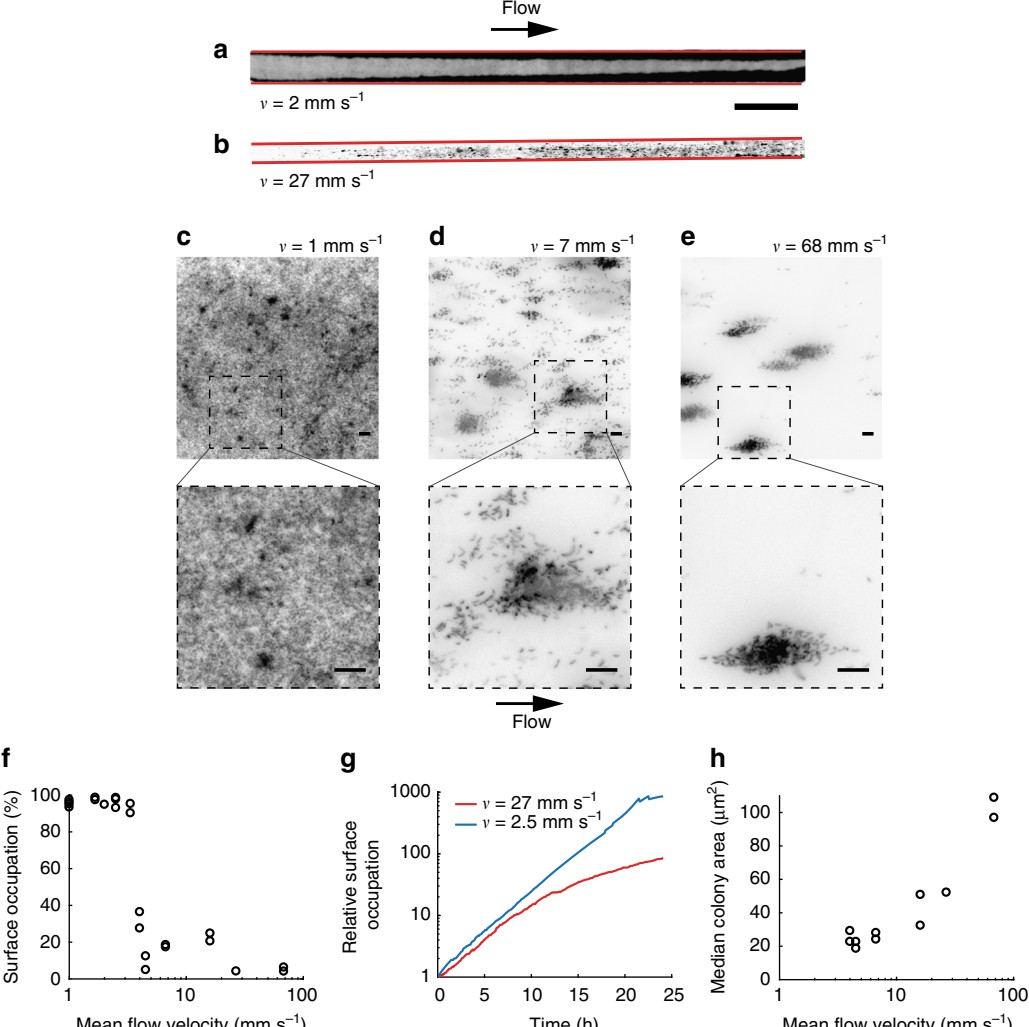

**Fig. 1** Flow modulates *C. crescentus* colonization patterns. **a**, **b** Top view, grayscale display of fluorescence microscopy images of *C. crescentus* after 48 h exposure to fluid flow in microchannels (cells are shown in black over a white background). In strong flow (**b**), biofilms grow into patterns of discrete cell clusters, unlike in weak flow (**a**). The edges of the microchannel are highlighted in red. Scale bar: 1 mm. **c**–**e** Colonization patterns at the channel centerline at three representative flow velocities, after 24 h of colonization under flow. In weaker flow (**c**), the channel surface is nearly saturated. At intermediate flow (**d**), multicellular clusters are surrounded by smaller groups or single isolated cells. In strong flow (**e**), biofilms grow mainly as multicellular clusters. Scale bars: 10 μm. **f**–**h** Fluid flow modulates kinetics and pattern geometry during surface colonization. **f** Surface occupation after 24 h of growth as a function of mean flow velocity. Each data point corresponds to an individual experiment. **g** Surface occupation over time for two representative flow velocities. **h** Median microcolony area after 24 h of growth as a function of mean flow velocity

after division is similar between WT and *flgE⁻*, suggesting that these distinct patterns are not a result of differences in adhesion between the two strains[20]. To further confirm that this difference in pattern formation between WT and *flgE⁻* is in fact due to motility, we compared the ability of WT and *flgE⁻* to attach to the surface in flow. We performed these attachment experiments using synchronized populations of WT and *flgE⁻* swarmer cells by connecting the outlet of a microchannel colonized with a 40-h-old biofilm to the inlet of a new microchannel. These measurements show that while WT cells are able to reach and attach to the surface at low flow, *flgE⁻* cells only rarely attach to the surface of the channel independently of flow velocity (Supplementary Fig. 1). Altogether, these observations demonstrate that the colonization patterns generated by *flgE⁻* are caused by a loss of swimming motility through a decrease in effective diffusion across the flow direction. Thus, motility plays a critical role in controlling surface occupation density and distribution.

**An advective-diffusion model for surface colonization.** Fluid flow transports bacteria directionally along streamlines, whereas cells swim in diffusive, Brownian-like trajectories in the absence of chemical gradients[26]. We, therefore, drew an analogy with advective-diffusion transport problems: the balance between flow-driven advective transport of single cells and their diffusive flagellar motility must contribute to the distinct colonization patterns observed in our experiments. We thus developed a scaling for the probability of attachment of a free-swimming bacterium as a function of fluid velocity by reasoning in terms of timescales. Fluid flow transports swarmer bacteria from their division site toward the channel outlet in a characteristic time $\tau_a = L/v$, where $L$ is the microchannel length and $v$ the mean flow velocity. During this time, a cell explores the depth of the channel by swimming, effectively diffusing in the direction perpendicular to the surface with characteristic timescale $\tau_D = h^2/D$, where $h$ is the channel height and $D$ the effective diffusion coefficient of a

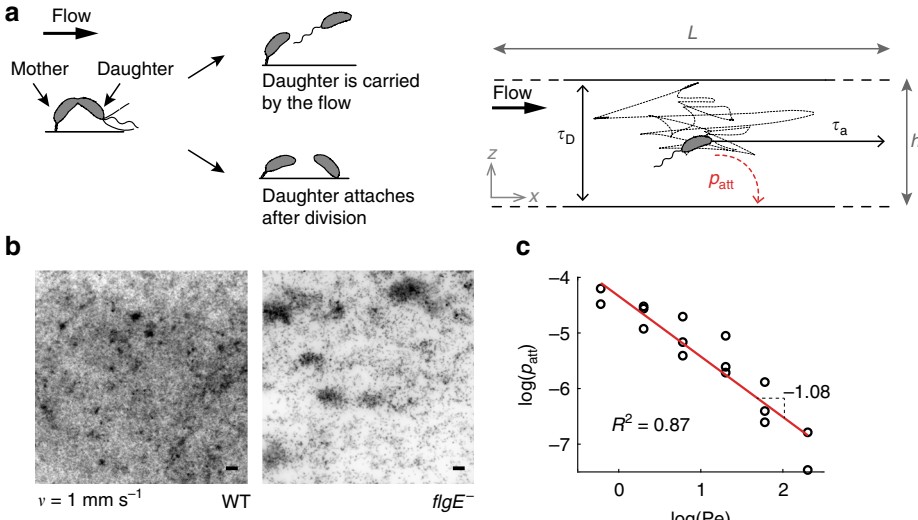

**Fig. 2** Physical mechanism for modulation of *C. crescentus* biofilm architecture. **a** *C. crescentus* divides asymmetrically: the mother cell is anchored to the surface and undergoing division. At the time of division, a daughter cell can either attach to the surface or be carried by the flow. If attachment occurs, the daughter immediately synthesizes a holdfast contributing to clonal expansion on the surface. If the daughter cell does not attach to the surface, it is subject to: (i) advective transport by fluid flow and (ii) diffusion-like transport generated by unbiased swimming. **b** Contribution of bacterial motility to surface colonization patterns. Fluorescence microscopy images of wild-type (WT) or flagellum-less (*flgE⁻*) *C. crescentus* after 24 h exposure to fluid flow (1 mm s⁻¹). Surface colonization by the *flgE⁻* mutant is slower and less saturated than WT. This qualitatively recapitulates the results observed in stronger flow for WT. Scale bars: 10 μm. **c** Attachment probability (attachment rate normalized by total bacterial flux) as a function of the Péclet number (Pe) on a logarithmic scale. A linear fit of the data indicates swarmer adhesion probability scales with Pe⁻¹, as suggested by our advective-diffusion model

bacterium attributed to unbiased swimming[26]. The probability that a free-swimming bacterium reattaches to the surface depends on the ratio of these two timescales: $\tau_D/\tau_a = (h^2v)/(DL)$, a nondimensional quantity resembling a Péclet number (Pe), which measures the relative contributions of advective to diffusive transport[27,28]. At large Pe ($\tau_a \ll \tau_D$) cells are rapidly washed out of the channel before encountering the surface so that the probability of attachment is low. In contrast, at very low Pe, diffusion dominates over flow; a planktonic cell has sufficient time to reach the surface before being flushed out of the channel, and may eventually reattach to the surface away from its stalked parent (high attachment probability). To validate this scaling, we flowed planktonic cells in microchannels and measured the attachment rate of WT cells as a function of applied flow velocity, effectively tuning Pe. We counted the number of cells attaching onto the surface per unit time, and estimated the corresponding $p_{att}$ by normalizing the rate of attachment with the incoming flux of cells. We found that attachment probability scales with Pe⁻¹ (Fig. 2c), which is consistent with the advection-diffusion model, validating our physical explanation of surface colonization patterns and rates.

**Flow modulates clonal lineage structure**. While patterns of surface colonization are crucial for initiating biofilm growth, they can also set the foundation for clonal lineage structure, a key factor influencing the evolution of microbial interaction traits[6]. The mechanisms by which environmental conditions, such as fluid flow, and microbial response to these factors influence the spatial architecture of polymicrobial communities, however, are still unclear. In mass transport phenomena, the balance between advective and diffusive transport strongly influences mixing of fluids and solutes[28]. By analogy, we reasoned that since surface occupation by *C. crescentus* is governed by advective diffusion, flow may also impact the mixing of distinct cell lineages and their social interactions. We grew *C. crescentus* biofilms in various flow conditions, starting from a one-to-one mixture of strains

constitutively expressing mKate or Venus fluorescent proteins whose doubling times are identical[20] (Fig. 3a). Consistent with advective-diffusion transport, surface populations of mKate- and Venus-expressing cells were well mixed in weak flow. There was no clear region where clonal lineages were segregated at scales larger than 10 μm. Within seemingly homogeneous clonal groups of cells, we could generally find invaders expressing the other fluorescent protein. At higher flow velocity, clonal groups were larger and segregated from each other, suggesting that they originated from a single parent cell.

The distribution of cross-lineage colony distances effectively measures segregation and thus strongly depends on flow intensity: in intermediate flow, all colonies expressing a given fluorescent protein are at most ~20 μm away from their nearest counterpart (Fig. 3b). The distribution is heavily weighted at low values of nearest neighbor distance (standard deviation = 3.5 μm). In contrast, at high flow intensity, the distribution of cross-lineage colony distances broadens dramatically (standard deviation = 12.8 μm). Colonies from each color variant can be separated by as much as 50 μm, and there is a substantial decrease in the frequency of small intercolony distances. This shift in distribution occurs progressively as flow intensity increases: the mean cross-lineage distance indeed increases as a function of mean flow velocity, demonstrating that segregation strengthens with flow (Fig. 3c). We hypothesized that, at low Pe, motility drives diffusive swimming trajectories to promote clonal mixing. We confirmed this by observing a reduction of clonal mixing of flagellum-less mutant at low flow intensity compared to WT (Fig. 4a). Consistent with this, in competition experiments between WT and *flgE⁻*, WT cells commonly invaded the biofilms of the nonmotile mutant, but the opposite was very rare (Supplementary Fig. 2). At high Pe, flow represses mixing of clones by carrying planktonic cells far from their parent, so that the spatial arrangement of WT and *flgE⁻* are similar (Fig. 4b). Together, these observations are consistent with a model where the balance between advection and diffusion of planktonic cells and deposition of daughter cells adjacent to their points of origin

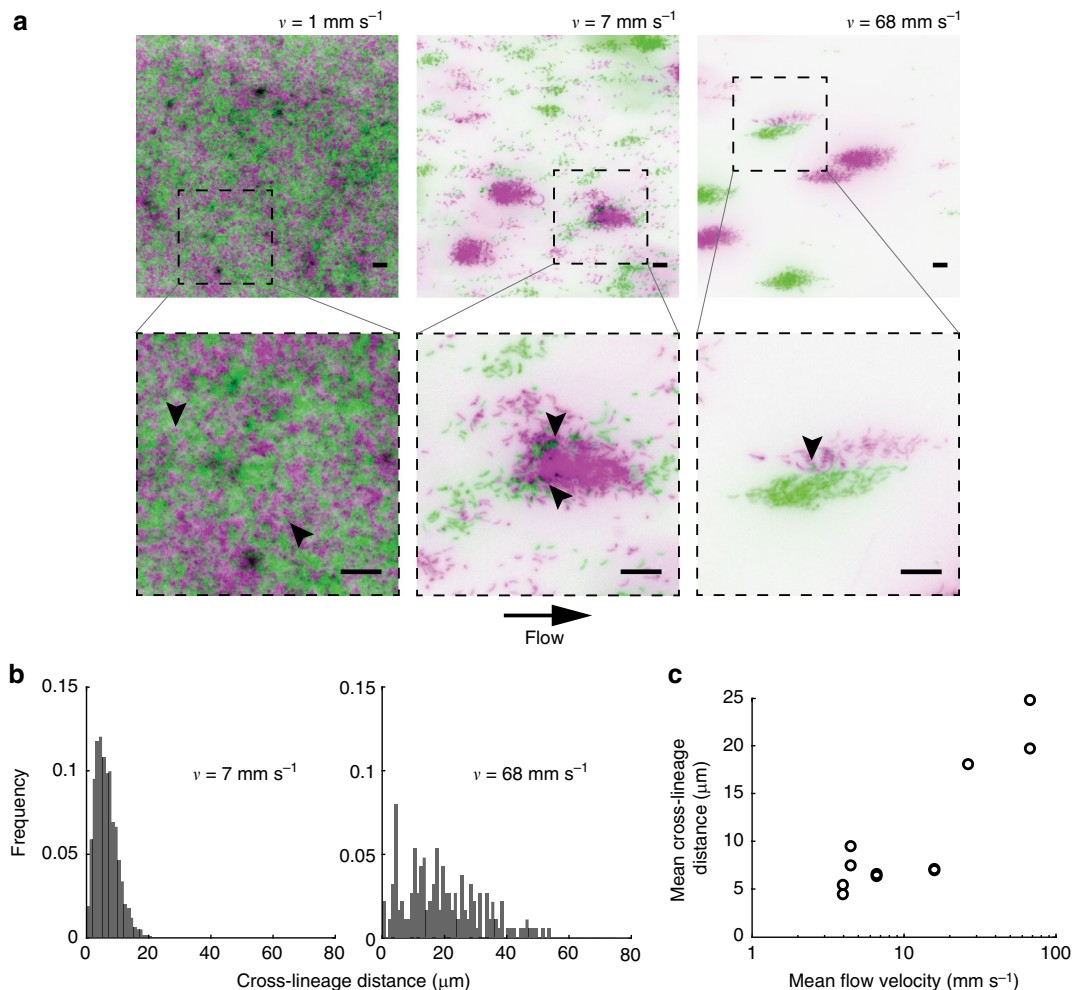

**Fig. 3** Flow modulates clonal structuring of *C. crescentus* biofilms. **a** Fluorescence microscopy images of *C. crescentus* biofilms (24 h). Two populations at equal density, expressing either mKate or Venus fluorescent proteins, were initially loaded in microchannels. The bottom row of images highlights the presence of invading cells (indicated by black arrowheads) within otherwise clonal clusters. Green: *C. crescentus* mKate. Magenta: *C. crescentus* Venus. Scale bars: 10 μm. **b** Distribution of cross-lineage colony distances (i.e., distance between green colonies and their nearest magenta neighbor, and vice versa) for two representative mean flow velocities (7 and 68 mm s$^{-1}$). The distribution broadens as flow velocity increases. **c** Cross-lineage colony distance, which can be used as a measure of clonal segregation, as a function of mean flow velocity. As flow velocity increases, the mean cross-lineage distance increases, indicating that biofilm mixing decreases

dictates the level of clonal structure within nascent *C. crescentus* biofilms (Fig. 4c). Finally, we verified that clonal patterns are conserved later in the colonization process, demonstrating that the dependence of biofilm spatial structure on flow is not a sole consequence of differences in colonization kinetics. After 6 days of growth, biofilms in both weak and strong flow regimes covered the surface entirely and extended into the channel depth. These mature biofilms retained the clonal structure set by the initial patterns of surface occupation we observed after 24 h of growth: the spatial distribution of clones remained highly mixed at low flow and relatively segregated at high flow (Fig. 5a). These differences in cellular arrangement are retained in the third dimension, as clonal clusters observed in strong flow essentially propagate as they grow normally to the channel surface (Fig. 5b).

## Discussion

Biofilms are permanently subject to the physical forces generated in their microenvironments. In their aquatic lifestyle, they must in particular cope with viscous forces induced by fluid flow. These forces strongly impact many elements of biofilm structure, including their overall morphology[29,30], solute flux into and out of the population[31] and detachment of cell clusters from surfaces[10]. Given the importance of hydrodynamic effects in remodeling biofilms and in transporting planktonic cells or aggregates[12], we anticipated that these forces also modulate spatial organization of surface associated bacterial collectives on many scales.

We demonstrated that the multi-scale feedbacks between surface attachment, daughter cell deposition, fluid transport, and dispersion by diffusion exert a strong influence on the morphological, spatial and genetic structure of biofilm populations. The early stages of surface colonization can set the foundations of subsequent biofilm architecture, influencing the spatial distributions of different strains and species, and the community's interaction networks. One critical ingredient to this process is probabilistic local attachment versus planktonic release of daughter cells. Any species in which extracellular matrix secretion has some likelihood of locally trapping recently divided daughter cells should display similar dynamics. Consistent with this, Martinez-Garcia et al.[18] recently showed using theory and experiments with *Vibrio cholerae* that flow, initial population

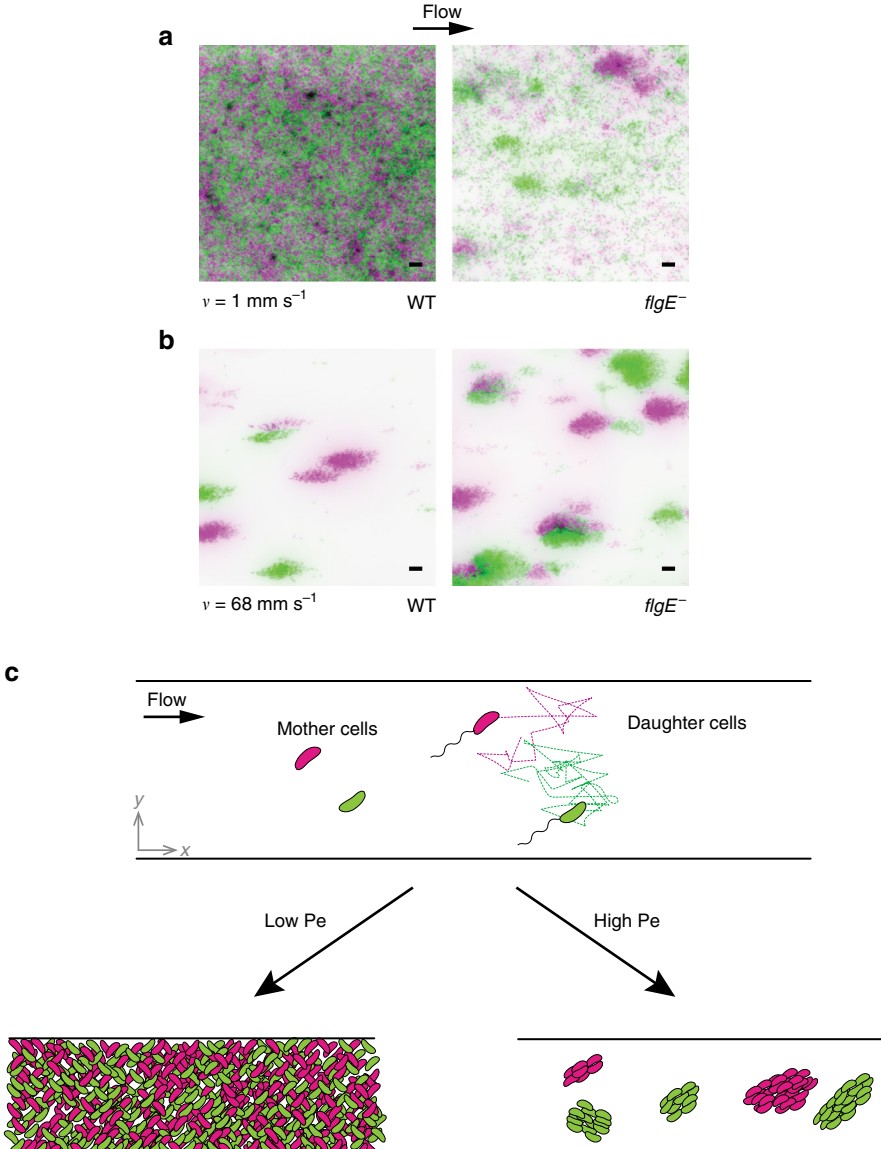

**Fig. 4** Impact of motility on the spatial structure of cell lineages. Fluorescence microscopy images of wild-type (WT) or flagellum-less (*flgE⁻*) *C. crescentus* after 24 h of growth in a microchannel under low (1 mm s⁻¹) or high (68 mm s⁻¹) flow. Two populations at equal density, expressing either mKate or Venus fluorescent proteins, were initially loaded in microchannels. **a** At low flow, the colonization pattern of *flgE⁻* shows lower surface coverage and larger cluster size than WT. This qualitatively recapitulates the results observed in stronger flow for WT, indicating that flagellum-powered swimming motility contributes to clonal dispersion. **b** Under strong flows, when advective transport dominates over bacterial swimming, WT and *flgE⁻* cells form similar biofilm patterns. Green: *C. crescentus* mKate. Magenta: *C. crescentus* Venus. Scale bars: 10 μm. **c** Within the framework of our model, the ratio of diffusive to advective transport timescales influences heterogeneity of clonal distributions in space. Indeed, Brownian-like single cell trajectories generate dispersion of bacterial clones across the channel surface at low Pe. Strong flows (large Pe) mitigate this effect by increasing clonality

density, and matrix secretion interact strongly to influence clonal colony size in early biofilm growth.

Even in the absence of matrix-mediated daughter cell surface attachment, asymmetries in adhesive properties may very likely appear between two daughter cells that are dividing symmetrically[32]. For example, a memory effect in *Pseudomonas aeruginosa* yields strong differences in the adhesive behavior of two sessile daughter cells, nearly recapitulating the pattern of *C. crescentus*, despite the absence of obvious cellular asymmetry[15,33,34]. Furthermore, the distribution of messenger molecules that regulate the production of matrix components is asymmetric in daughter *P. aeruginosa* cells, potentially differentially affecting adhesive properties[15,33,34]. The same effect may also arise in *Escherichia coli*[32]. The balance between directional advective and random diffusive trajectories of detached planktonic cells constitutes a second ingredient setting the spatial structure of biofilm communities. In the same manner as transport of particulate matter, a Péclet number can be used to predict the emergence of motility- and flow-induced morphological transitions. Advective-diffusion has been employed to model the dispersion of planktonic bacteria in laminar flow[35], and even the dispersion of airborne plant seeds[36]. Likewise, phase-like transitions between multicellular phases in *Myxococcus xanthus* can be described using a Péclet number quantifying the relative contribution of directional cell displacement to rotational diffusion[37].

The distinction between spatial segregation versus mixture of distinct clonal lineages is of key importance to the expected evolutionary trajectories of numerous traits, and in particular,

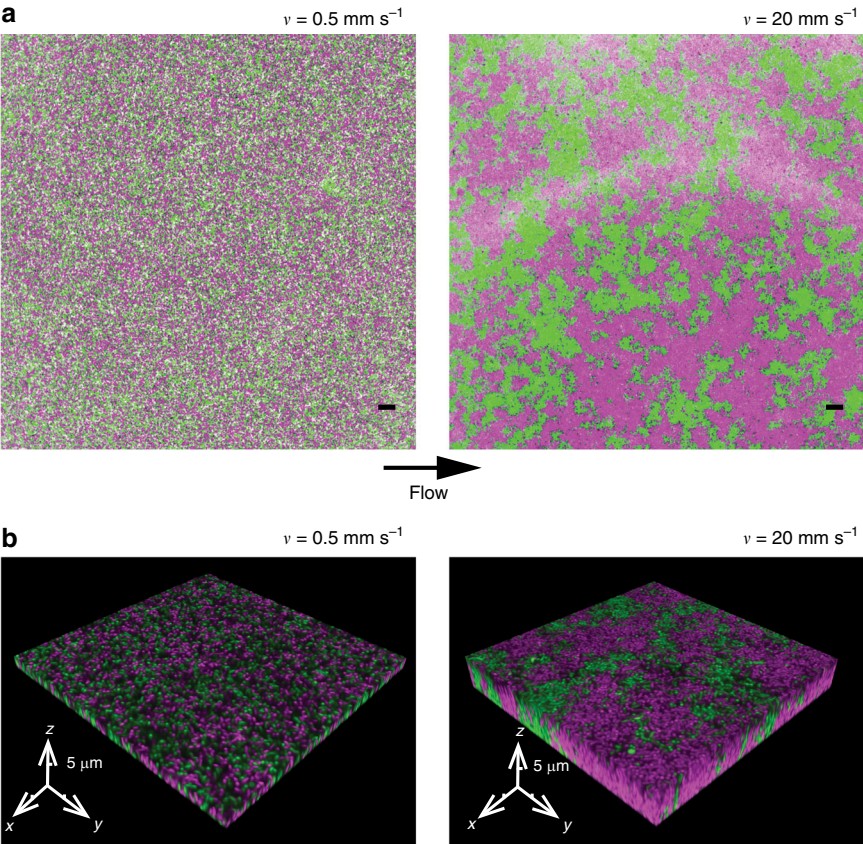

**Fig. 5** Flow-dependent colonization patterns persist on longer time scales and in three dimensions. Biofilms of *C. crescentus* expressing either mKate or Venus fluorescent proteins, grown under low (0.5 mm s$^{-1}$) or high (20 mm s$^{-1}$) flow during 6 days. **a** At the channel surface, biofilm mixing patterns after 6 days recapitulate those obtained after 24 h (Fig. 3a). Scale bar: 10 μm. **b** 3D rendering of a z-stack acquisition of biofilms grown for 6 days under flow. Characteristic mixing patterns previously observed on the surface of channel also extend in the third dimension. Green: *C. crescentus* mKate. Magenta: *C. crescentus* Venus

traits that exert a fitness impact on nearby cells. These include all contact-mediated interactions as well as helpful or harmful interactions mediated by compounds secreted into the extracellular space. The spatial orientation of genetic lineages and the movement of these compounds, which is also controlled by local fluid transport regime, interact to determine whether these interactive behaviors are evolutionarily stable[6]. Tight clustering of clonemates as occurs under strong flow, for instance, increases the evolutionary stability of locally cooperative phenotypes. On the other hand, antagonistic traits operate most effectively under mixed lineage conditions, when the targets of the harmful phenotype are easily within reach. Our study highlights the importance of the mechanical environment in shaping the foundation of biofilm community architecture at single cell and cell collective length scales. Our model system consisting of *C. crescentus* colonizing flat surfaces in unidirectional flow is minimalistic, but generates insights for the understanding of organization of more complex communities in more intricate mechanical environments. An important future direction will be to recapitulate more ecologically realistic conditions, to understand how flow structure shapes spatial organization of biofilms in environments such as the widely varying host mucosa and rhizosphere[38,39].

## Methods

**Design and fabrication of the microfluidic chips**. We fabricated the microfluidic chips following standard soft lithography techniques. More specifically, for the 24- and 48-h long biofilm experiments, we designed 1-cm long, 500 or 250-μm wide channels in Autodesk AutoCAD 2018 and printed them on a soft plastic photomask. We then coated silicon wafers with photoresist (SU8 2025, Microchem), with varying thicknesses (25, 50, and 90 μm) to allow a wider range of mean flow velocities for identical flow rate settings. The wafer was exposed to UV light through the mask and developed in PGMEA (Sigma-Aldrich) in order to produce a mold. PDMS (Sylgard 184, Dow Corning) was subsequently casted on the mold and cured at 80 °C for about 1 h 30 min. After cutting out the chips, we punched 1 mm inlet and outlet ports. We finally bonded the PDMS chips to glass coverslips (Marienfeld 1.5) in a ZEPTO plasma cleaner (Diener electronic). To fabricate channels for the 6-day long biofilm experiments, we followed a similar procedure, but adjusted the dimensions of the channel to leave more space for large 3D structures to form. More precisely, the channel was 2 mm wide, 110 μm high.

**Bacterial strains**. We used *C. crescentus* strains CB15 constitutively expressing chromosomally integrated fluorescent protein genes *Venus* or *mKate* off a modified *lac* promoter[20]. These strains were grown in peptone yeast extract (PYE) medium supplemented with 5 μg/ml of kanamycin (PYE-Kan) in a shaking incubator set to 30 °C. For the experiments involving nonmotile CB15, we inserted either mKate or Venus in the chromosome of the flagellum-less mutant CB15 *flgE*$^{-}$ using plasmids pXGFPC-2 Plac::mKate2 and pXGFPC-2 Plac::Venus, respectively[20]. We prepared electrocompetent CB15 *flgE*$^{-}$ by centrifuging 3 ml of stationary phase culture and rinsing it two times with cold Milli-Q water (Merck Millipore). About 600 ng of plasmid were added for transformation and the bacteria were then plated on PYE-Kan plates.

**Biofilm growth in microfluidic chambers**. At the start of every experiment, the bacterial cultures had an optical density of approximately 0.15 (~4.5 × 10$^{8}$ CFU ml$^{-1}$). Equal volumes of CB15 mKate and CB15 Venus were diluted in PYE-Kan to a final 1:10 concentration. We then loaded the bacterial mixture in a microchannel using a micropipette, and let them adhere for 3 min (WT) or 15 min (*flgE*$^{-}$) before washing the channel with PYE-Kan. For all conditions but the highest flow velocity ($v = 68$ mm s$^{-1}$ for 24 h biofilms, and $v = 20$ mm s$^{-1}$ for 6-day biofilms), we connected the inlet port to a disposable PYE-Kan-filled syringe (BD Plastipak) using a 1.09 mm outer diameter polyethylene tube (Instech) and a 27G needle (Instech). The syringe was then mounted onto a syringe pump (ZS100, ChuangRui Pump). For the highest flow conditions ($v = 68$ mm s$^{-1}$ for 24 h biofilms, and

$v = 20$ mm s$^{-1}$ for 6-day biofilms), we connected the inlet port to a PYE-Kan-filled beaker via two imbricated tubes (polyethylene tubing as described above, and Tygon-LFL tubes with an inner diameter of 0.76 mm (Ismatec)). We mounted the setup onto a peristaltic pump (MCP, Ismatec) allowing us to work with larger volumes than the syringe pump. For every experiment, we connected the outlet port to a waste container using polyethylene tubing. We finally placed the chip in a 30 °C incubator and applied a controlled flow of PYE-Kan to the microchannels for 24 h, 48 h, or 6 days depending on the experiment. The mean flow velocity ($v$) was calculated from the selected flow rate ($Q$) and channel cross-sectional area ($A$) as such $v = Q/A$.

**Visualization.** For all visualizations of biofilms grown up to 48 h, we used a Nikon TiE epifluorescence microscope equipped with a Hamamatsu ORCA Flash 4 camera and a 40× Plan APO NA 0.9 objective. The full-channel images were stitched using the NIS-Elements software. All single cell level pictures presented in this work were taken 9 mm away downstream of the inlet. For the timelapse experiments (Supplementary Movies 1 and 2), we acquired images every 5 min for 24 h. To visualize 6 day old biofilms, we used a Leica SP8 confocal microscope equipped with a white laser, a 25× HC FLUOTAR NA 0.95, water-immersion objective, as well as a 63× HC PL APO NA 1.40 oil-immersion objective for high magnification z-stack acquisitions. We used Imaris (Bitplane) for three-dimensional rendering of z-stack pictures (Fig. 5b).

**Data analysis.** Data analysis was conducted using Matlab (Mathworks). To discriminate cells from background, the images were segmented with the built-in Matlab function *imbinarize* using an adaptive threshold (*adaptthresh* built-in Matlab function), the sensitivity of which varied depending on the median intensity of the picture. Similarly, the percentage of background removed was also determined by the median intensity. Finally, we filtered out objects smaller than 15 pixels, since this value was observed to be the minimal area of a single cell standing vertically. After segmentation, pictures were visually assessed to ensure the quality of segmentation. In rare cases (4 pictures out of 50), segmentation was aberrant (i.e., the segmented features did not correspond to the bacteria in the raw picture, likely due to uneven background) and the images had to be excluded from the analysis.

To calculate the surface coverage and microcolony area, we merged the segmented pictures originating from mKate and Venus using the logical *or* function. To quantify surface coverage, we divided the area of black pixels (i.e., pixels containing a part of cell) by the total area of an image.

We observed that single cell clusters were difficult or even impossible to discriminate by eye when surface coverage was larger than 80%. Therefore, we only included segmented pictures with a surface coverage ≤80% for the measurement of microcolony area. We also filtered out any object smaller than 200 pixels, which approximately corresponds to a group of five cells (average cell size: 1.29 µm$^2$ ≈ 40.2 pixels, $N = 80$ cells). We then closed the pictures using a built-in Matlab function (with a disk structuring element having a radius of five pixels) and calculated the area of every colony. The median colony area was finally calculated for each image.

To quantify the degree of mixing of the biofilms, we again only studied segmented pictures with a surface coverage ≤80%. In addition, unlike for surface coverage and colony area quantification, we analyzed mKate and Venus pictures separately. We closed all the pictures as mentioned above. We then calculated the distance between the centroid of an object and its nearest neighbor expressing the other fluorescent protein, using the built-in Matlab function *knnsearch*. This operation was repeated for every object in every picture. Finally, the mean cross-lineage distance was calculated for each experimental condition, considering distances from both fluorescently labeled populations.

**Estimation of attachment probabilities.** To estimate the attachment probability of swarmer *C. crescentus* in different flow conditions, we flowed CB15 Venus cells in a 500-µm wide, 90-µm high microchannel using a syringe pump. The flow rates varied between 0.81 and 270 µl min$^{-1}$ (mean flow velocities from 0.3 to 100 mm s$^{-1}$, respectively). Each condition was repeated two to three times. Bacteria were visualized by fluorescence microscopy (one frame recorded every second during one minute) and single attachment events were counted. Bacteria had to remain on the surface for at least three consecutive frames at the same location to be counted as attached. The number of bacteria attached over time was plotted for each flow condition and, using a linear fit, we extracted the attachment rate from the slope of these curves. The attachment probability was then computed as follows:

$$p_{att} = \frac{r}{J \cdot A} = \frac{r}{Q \cdot C} \qquad (1)$$

where $r$ is the attachment rate, $C$ is the bacterial concentration, and $J$ is the bacterial flux, defined as $J = (Q \cdot C)/A$. Normalization by the bacterial flux is necessary, because the number of bacteria going through the channel during a given time depends on flow rate. Also note that the bacteria loaded in the channel contained a mixture of swarmer and stalked cells, thus our measurement of $p_{att}$ is underestimated compared to biofilm growth conditions where all planktonic cells are swarmers.

To determine the dependence of attachment on the flow regime, we plotted the logarithm of $p_{att}$ as a function of the logarithm of Pe. As defined above, Pe $= (h^2 v)/(DL)$; we assumed the diffusion coefficient of *C. crescentus* to be equal to that of *E. coli* reported before[26], namely $D = 4 \times 10^{-6}$ cm$^2$ s$^{-1}$. We finally determined the proportionality relation between $p_{att}$ and Pe from the slope of the curve, since $\alpha \log(x) = \log(x^\alpha)$.

For Supplementary Fig. 1, we generated a population of synchronized swarmer cells. We used a strategy where we connected a biofilm-containing microchannel shedding newly divided swarmers to a bare microchannel. We loaded a mixture of Venus-expressing CB15 WT and mKate-expressing CB15 *flgE*$^-$ cells into a microchannel (channel 1) and let them grow at 30 °C during 40 h under flow ($v = 7$ mm s$^{-1}$). We then connected the outlet of channel 1 to the inlet of another microchannel (channel 2) using polyethylene tubing. This way, swarmer cells released from the biofilm of channel 1 were transferred to the empty channel 2 at the applied flow rate. For each flow velocity, bacteria were imaged every 2 s for 2 min from which we quantified newly attached cells at every time frame. Finally, to determine the total concentration of cells, a CFU count was performed. We calculated the relative ratio of WT to mutant cells feeding channel 2 by microscopic observation of the effluent of channel 1. We then adjusted the obtained values for rates of attachment to account for the relative proportions of WT versus *flgE*$^-$ cells.

**Reporting summary.** Further information on research design is available in the Nature Research Reporting Summary linked to this article.

## Data availability
All data are available from the corresponding author upon reasonable request.

## Code availability
All codes are available from the corresponding author upon reasonable request.

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

## Acknowledgements

T.R. and A.P. are supported by the Swiss National Science Foundation, Projects grant 31003A_169377 and the Giorgio Cavaglieri Foundation. C.D.N. is supported by the National Science Foundation (MCB 1817342), a Burke Award from Dartmouth College, a pilot award from the Cystic Fibrosis Foundation (STANTO15RO), and NIH grant P20-GM113132 to the Dartmouth BioMT COBRE.

## Author contributions

T.R., C.N., and A.P. conceptualized the study, T.R. performed the experiments and data analysis. T.R., C.N., and A.P. wrote the paper.

## Additional information

**Competing interests:** The authors declare no competing interests.

