## [Peer Review File · Nature Communications]

Reviewer #1 (Remarks to the Author):

All my previous concerns and comments have been satisfactorily addressed, except for my main previous concern, that this result seems very obvious/intuitive. I certainly agree that there is a value in casting biofilm structure as the result of advection vs diffusion, and there is a value to the quantitative description provided here. What is not obvious to me is why the authors claim (in their response letter) that flow is not naturally/intuitively associated with mixing by microbiologists. I really don't understand this - here, flow is providing a method of transport that, combined with motility, allows different bacterial strains to mix. I don't see why a microbiologist would find this surprising.

However, science doesn't have to be surprising to be good science and I do like the physical analysis applied here - and I repeat that there is a value to formal, quantitative understanding of a phenomenon even when the qualitative effect is un-surprising, so I am fine with seeing this published.

Reviewer #2 (Remarks to the Author):

I find that the authors have properly addressed most of my concerns and I think the current manuscript will make a significant contribution to an important field.

I would still like my main concern from the previous version addressed more directly in the text. While I find that the experiment they show in Fig S1 adds evidence towards their contention that their flgE phenotype is mainly due to lack of motility, it remains a fact that this type of mutation has regulatory impacts. This should be mentioned, with proper references, so the readers can decide for themselves if they agree. In other words, the authors should at least discuss the possibility that their phenotype could be due holdfast or pili misregulation (mentioning Hug et al, Science 2017; Berne et al, Mol Micro 2018; Ellison et al J Bact, 2019; and Hershey et al mBio 2019), even if in the end they conclude that it is not, based on the results provided in fig S1 (and their previous work, ref 20).

Reviewer #3 (Remarks to the Author):

I have carefully read the revised version of the manuscript. All my comments have been addressed in a convincing way.

In my opinion, the authors have drawn a coherent picture of early biofilm formation under various flow conditions, that can serve as a solid ground to study ecological interactions and evolution in such a peculiar environment.

I believe this is a very interesting contribution, well suited for publication in Nature Communications.

Response to reviewers

Cellular advective-diffusion drives the emergence of bacterial surface colonization patterns and heterogeneity

Tamara Rossy, Carey D. Nadell, Alexandre Persat (corresponding author)

Reviewer Comments:

Reviewer #1, after revision:

All my previous concerns and comments have been satisfactorially addressed, except for my main previous concern, that this result seems very obvious/intuitive. I certainly agree that there is a value in casting biofilm structure as the result of advection vs diffusion, and there is a value to the quantitative description provided here. What is not obvious to me is why the authors claim (in their response letter) that flow is not naturally/intuitively associated with mixing by microbiologists. I really don't understand this - here, flow is providing a method of transport that, combined with motility, allows different bacterial strains to mix. I don't see why a microbiologist would find this surprising.

However, science doesn't have to be surprising to be good science and I do like the physical analysis applied here - and I repeat that there is a value to formal, quantitative understanding of a phenomenon even when the qualitative effect is un-surprising, so I am fine with seeing this published.

We thank the reviewer for his/her enthusiasm. We would like to point out that we heard directly from microbiologists (through conversation and questions at seminars, so we agree that this might not be the

norm) that our results were counterintuitive. We believe that one may associate strong flow to turbulence and consequently to mixing.

Reviewer #2, after revision:

I find that the authors have properly addressed most of my concerns and I think the current manuscript will make a significant contribution to an important field.

*I would still like my main concern from the previous version addressed more directly in the text. While I find that the experiment they show in Fig S1 adds evidence towards their contention that their *flgE* phenotype is mainly due to lack of motility, it remains a fact that this type of mutation has regulatory impacts. This should be mentioned, with proper references, so the readers can decide for themselves if they agree. In other words, the authors should at least discuss the possibility that their phenotype could be due holdfast or pili misregulation (mentioning Hug et al, Science 2017; Berne et al, Mol Micro 2018; Ellison et al J Bact, 2019; and Hershey et al mBio 2019), even if in the end they conclude that it is not, based on the results provided in fig S1 (and their previous work, ref 20).*

We agree with the reviewer and have now modified the final manuscript to discuss this and added some of the suggested references (lines 110-113).

Reviewer 3, after revision:

I have carefully read the revised version of the manuscript. All my comments have been addressed in a convincing way.

In my opinion, the authors have drawn a coherent picture of early biofilm formation under various

flow conditions, that can serve as a solid ground to study ecological interactions and evolution in such a peculiar environment.

I believe this is a very interesting contribution, well suited for publication in Nature Communications.

We are glad to read that reviewer #3 finds our work convincing and we thank him/her for these comments.